# Three novel bird strike likelihood modelling techniques: The case of Brisbane Airport, Australia

**Robert Andrews**[1]*, **Bayan Bevrani**[1], **Brigitte Colin**[1], **Moe T. Wynn**[1], **Arthur H. M. ter Hofstede**[1], **Jackson Ring**[2]

**1** School of Information Systems, Faculty of Science, Queensland University of Technology, Brisbane, Queensland, Australia, **2** Wildlife Management Group, Brisbane Airport Corporation, Brisbane, Queensland, Australia

* r.andrews@qut.edu.au

**Data Availability Statement:** All relevant data are within the paper and its Supporting information files. A GitHub repository has also been created to make larger files available to reviewers and readers

## Abstract

The risk posed by wildlife to air transportation is of great concern worldwide. In Australia alone, 17,336 bird-strike incidents and 401 animal-strike incidents were reported to the Air Transport Safety Board (ATSB) in the period 2010-2019. Moreover, when collisions do occur, the impact can be catastrophic (loss of life, loss of aircraft) and involve significant cost to the affected airline and airport operator (estimated at globally US$1.2 billion per year). On the other side of the coin, civil aviation, and airport operations have significantly affected bird populations. There has been an increasing number of bird strikes, generally fatal to individual birds involved, reported worldwide (annual average of 12,219 reported strikes between 2008-2015 being nearly double the annual average of 6,702 strikes reported 2001-2007) (ICAO, 2018). Airport operations including construction of airport infrastructure, frequent take-offs and landings, airport noise and lights, and wildlife hazard management practices aimed at reducing risk of birdstrike, e.g., spraying to remove weeds and invertebrates, drainage, and even direct killing of individual hazard species, may result in habitat fragmentation, population decline, and rare bird extinction adjacent to airports (Kelly T, 2006; Zhao B, 2019; Steele WK, 2021). Nevertheless, there remains an imperative to continually improve wildlife hazard management methods and strategies so as to reduce the risk to aircraft and to bird populations. Current approved wildlife risk assessment techniques in Australia are limited to ranking of identified hazard species, i.e., are 'static' and, as such, do not provide a day-to-day risk/collision likelihood. The purpose of this study is to move towards a dynamic, evidence-based risk assessment model of wildlife hazards at airports. Ideally, such a model should be sufficiently sensitive and responsive to changing environmental conditions to be able to inform both short and longer term risk mitigation decisions. Challenges include the identification and quantification of contributory risk factors, and the selection and configuration of modelling technique(s) that meet the aforementioned requirements. In this article we focus on likelihood of bird strike and introduce three distinct, but complementary, assessment techniques, i.e., **A**lgebraic, **B**ayesian, and **C**lustering (ABC) for measuring the likelihood of bird strike in the face of constantly changing environmental conditions. The **ABC** techniques are evaluated using environment

https://github.com/robertandrews59/ABCBirdstrike".

**Funding:** The grant funding was used for (partial) salaries of the following authors (as QUT employees): BB, BC, RA. The funder provided support in the form of salaries for author JR (as an employee of BAC). The specific roles of these authors are articulated in the 'author contributions' section. The funder did not have any role in the analysis, or preparation of manuscript. The funder was involved in validating the study design, and made historical data collected by its Wildlife Hazard Management team available to the researchers. The funder was also involved in the decision to publish.

**Competing interests:** The authors have read the journal's policy and have the following competing interest: JR was, at the time of writing, an employee of Brisbane Airport Corporation. This does not alter our adherence to PLOS ONE policies on sharing data and materials. There are no patents, products in development or marketed products associated with this research to declare.

and wildlife observations routinely collected by the Brisbane Airport Corporation (**BAC**) wildlife hazard management team. Results indicate that each of the techniques meet the requirements of providing dynamic, realistic collision risks in the face of changing environmental conditions.

## Introduction

The risk posed by wildlife to air transportation is of great concern worldwide. Wildlife Hazard Management (WHM) is a mandatory procedure in every significant international and national airport [1]. A hazard is defined as "the condition or circumstance that could lead to damage or destruction of an aircraft, or to loss of life as a result of aircraft operations" [2]. Statistics and data on wildlife collisions with aircraft give some indication of the scale of the problem. In Australia alone, 17,336 bird-strikes occurrences and 401 animal-strike incidents were reported to the Australian Transport Safety Bureau between 2010 and 2019 [3] and 17,360 strikes reported to the US Federal Aviation Authority (FAA) in a single year (2019) [4]. Wildlife strikes can have catastrophic consequences. The FAA report more than 301 people killed and over 298 aircraft destroyed globally as a consequence of wildlife strikes since 1988 [5]. It is estimated that wildlife strikes incur cost to the civil aviation industry in the US of approximately $625 million per year [6] and globally US$1.2 billion per year [7]. On the other side of the coin, civil aviation, and airport operations have significantly affected bird populations. There has been an increasing number of bird strikes, generally fatal to individual birds involved, reported worldwide (annual average of 12,219 reported strikes between 2008–2015 being nearly double the annual average of 6,702 strikes reported 2001–2007) [8]. Airport operations including construction of airport infrastructure, frequent take-offs and landings, airport noise and lights, and wildlife hazard management practices aimed at reducing risk of birdstrike, e.g., spraying to remove weeds and invertebrates, drainage, and even direct killing of individual hazard species, may result in habitat fragmentation, population decline, and rare bird extinction adjacent to airports [9–11]. Nevertheless, there remains an imperative to continually improve wildlife hazard management methods and strategies so as to reduce the risk to aircraft and to bird populations.

Certification or accreditation of an airport with the regulatory authority is contingent on the airport implementing a Wildlife Hazard Management Plan (WHMP) for the management of wildlife hazards. In Australia, compliance with the Civil Aviation Safety Authority (CASA's) Advisory Circular AC 139–26(0) Wildlife Hazard Management at Aerodromes [12] requires of airport operators, that when assessing wildlife hazard risk, "individual species should be identified and prioritised in order of risk" [12]. AC 139–26(0) also points out the airport operator's requirement to provide advice to airmen at varying levels of immediacy including (i) standing caution for a bird or animal hazard that poses a constant risk, (ii) periodic when there is a significantly increased risk, for a relatively constrained period of time, posed by a hazard species, and (iii) immediate advice of risk from the control tower to warn approaching and departing aircraft of, say, birds (or bats), detected in the flight path.

Essential components of a WHMP are risk assessment techniques or tools that (i) comply with regulations and, (ii) are sensitive enough to properly align risk with changing conditions around the airfield. Such a management plan (risk mitigation strategy) will incorporate reactive elements (such as providing warnings to pilots as they approach or depart from the airport regarding the presence of one or more hazard species) and proactive elements (such as altering

habitat in the vicinity of the airport with a view to reducing hazard species' populations over time).

Various wildlife hazard risk assessment approaches have been proposed/implemented around the world. Common to all, is the aim of reducing the likelihood of birdstrike, however differences in approaches characterised by their (i) sphere of application (local airport to (inter-)national), (ii) forecast horizon (static to real-time), and (iii) realisation (as being technique/algorithm, a model, or an implemented system). Here we take the definitions of technique (theoretical, mathematically based structure), model (framework of techniques and algorithms to describe/predict real word situation), and system (interacting components including models, sensors, communications networks to utlise model outputs) from [13]. The following models/systems are at large spatial scale, i.e., larger scale than individual airport. The models/systems differ also in their forecast horizon. For example, real-time birdstrike warning systems include the Avian Hazard Advisory System (AHAS) [14] in the United States, the Dutch Radar Observation of Bird Intensities system (Dutch ROBIN system—https://www.robinradar.com/) in the Netherlands. In Europe, the FlySafe Bird Avoidance Model [15] provides near real-time information and forecast on large scale bird movement in the air space of The Netherlands, Germany and Belgium. Longer forecast horizon model/systems include the United States and North America Bird Avoidance Model (USBAM) [16], the German birdstrike risk forecast model [13], and the Swiss/Dutch Dynamic Bird Migration Model [13, 17]. The USBAM is a statistical model, while the German birdstrike risk forecast model is a conceptual model, and the Swiss/Dutch dynamic bird migration model is a simulation model. Purely static, wildlife species risk ranking techniques/algorithms include Allan's formula [11], Paton's approach [18], and Carter's formula [19] These techniques are intended for application at individual airports.

Little attention has been paid to applying machine learning to aspects of birdstrike risk. Rosa et al. [20] applied 6 different machine learning techniques to data collected by marine radar in Portugal. This study concluded that all techniques were able to distinguish between birds and stationary objects, but distinguishing between different species of birds was less successful. Recently, Nimmagadda et al. [21] applied decision tree and naive Bayesian approaches to the problem of predicting whether an airline crash has occurred due to a birdstrike. Verma et al. [22] provides a literature review of various techniques for prediction of general aviation accidents including, among others, Bayesian Networks, Artificial Neural Networks, data mining, and ensemble approaches. None of these techniques were specific to predicting likelihood of wildlife collision on/near an airfield.

Risk assessment techniques/methods generally integrate the likelihood of occurrence of a risk event with the severity of an event (or any other risk-related factors) to give an overall risk rating. Examples of such techniques can be found in [19, 23, 24].

Allan's formula [7], endorsed by the Australian Airports Association for use in Australian airports [25], assesses hazard species' risk by combining the probability of the species being involved in a strike event (determined using only historical strike data) with the estimated severity of a strike (calculated from the average mass of an adult member of the hazard (bird) species adjusted for flocking behaviour). The principal function of Allan's approach is to identify and rank hazard species. While being easy to implement in practice, Allan's approach has limitations as a day-to-day indicator of risk due to wildlife including (i) any species has a zero probability of involvement in a collision (and hence a low risk) until a strike is actually recorded, and (ii) the risk is largely static since it is derived from a rolling window of the past 5 years' observations, i.e. it does not take immediate conditions or short-term changes or trends into account.

Similar to Allan's formula, Paton's approach [18] is data-driven and uses a severity/likelihood matrix to assign a risk rating to a species (with however, significant differences to Allan's formula in the way in which the severity and likelihood categories are calculated). In particular, the likelihood indicators used in Paton's approach acknowledge that conditions on the airfield vary over time (e.g. number of birds on the airfield), making it sensitive to changing environmental conditions and potentially useful as the basis for a dynamic risk model.

It should be noted that the principal use of both Allan's approach and Paton's approach is in identifying and ranking hazard species. As such, both Allan's formula and Paton's approach are accepted by Australia's Civil Aviation Safety Authority (CASA). *Neither method however, is suitable for providing a risk assessment directly related to current conditions at the airport.*

Further, static risk ranking approaches do not take into account seasonal or environmental factors, and hence, cannot be used to (i) reason about likely future states on the airfield, or (ii) to plan mitigation measures in anticipation of some likely future states. By contrast, the techniques developed in this study do provide evidence-based, dynamic, collision likelihood assessments, and can be used to inform wildlife hazard managers of likely, required, mitigation activities. Here we note that risk assessment also includes taking into account a severity measure. This aspect was outside the scope of this paper and we concern ourselves with only collision likelihood.

## Materials and methods

### Study area and data collection

To develop these methods, a case study was conducted at the Brisbane International Airport, Australia located at the geographic coordinates 27˚23'07.58" S, 153˚07'13.48" E and an elevation of -3 m below mean sea level. It is located adjacent to the Brisbane coastline in a temperate (mesothermal) climate zone, defined as humid subtropical climate (Cfa) and temperate oceanic climate (Cfb) according to the Koeppen climate classification [26] system. The climate is characterised by hot and warm summers, without a dry season, although the average precipitation is higher in summer months than in winter months. Average temperatures range between 8˚C in the coldest month (July) to 27˚C in summer months. Within a 3km radius of the Brisbane airport natural features and wildlife attractants include the port of Brisbane, an oil refinery, refuse dumping ground, waste transfer station, a major golf club, big cemetery, wetlands reserve, waste water treatment plant, the Kedron Brook Floodway and several smaller brooks and creeks.

The (routinely collected) environmental, geographical and wildlife data used in this paper was provided by Brisbane Airport Corporation (BAC) in Australia and covers the period 1 May 2017 to 30 June 2019. The selection of variables was made in collaboration with our industry partner (BAC) and incorporated the expert knowledge of BAC's wildlife management team. The data set comprised six main data aspects: (i) environmental data such as daily rainfall and temperature, (ii) existing infrastructure features such as runways, taxi ways, terminal buildings, etc., (iii) a proximity weighting factor for zones on the airfield (relative to the distance of the zone from areas of high-speed aircraft movement such as runways where strikes are most frequently reported), (iv) details of reported bird strikes during the 761 days (1 May 2017 to 30 June 2019), (v) daily counts of three bird species nominated as hazard species of interest by BAC, and (vi) details of mitigation (harassment) strategies conducted to reduce the number of birds adjacent to aircraft movement corridors.

Wildlife observations are carried out daily between 06:00 and 09:00 by the WHM team at BAC. The count is conducted by a single team member, (usually) traversing the airport on a standard route. The airport is divided into 11 zones. The daily count data records the date and

time of wildlife observations, the species observed and the number of individuals of the species, as well as the location on the airport (zone) where the species was observed. Environmental variables such as rainfall, wind speed, wind direction, cloud height and cover are also recorded. Wildlife 'harassment' activities are also recorded (harassment is moving hazard species away from runways and taxi-ways). The date, time, and method (e.g. siren, pyrotechnics, etc.) of harassment activities are recorded along with the species and number of individuals harassed. Lastly, details of strikes are recorded including date and time, species and number involved, location (zone), aircraft, damage to aircraft, and whether the strike can be confirmed (by remains on the ground or evidence on the aircraft). S1 Table summarises the strike information for the three hazard species involved in the study during the period of the study.

BAC nominated the three hazard species it considered as being high risk species for inclusion in the study. The Cattle Egret (*Bubulcus ibis*) and Nankeen Kestrel (*Falco cenchroides*) were included due to their comparatively frequent involvement in collisions. The Straw-necked Ibis (*Threskiornis spinicollis*) was included due to the large numbers of this species (flocks of thousands of individuals) frequently observed on the airfield. The data set was a rectangular array—761 rows for each of three bird species, comprising 80 explanatory variables as a mix of continuous real values, categorical attributes and ordinal data.

Discussions with the BAC WHM team added some context to some relevant behaviours of the study hazard species.

- Nankeen Kestrels are small, non-migratory, raptors (preying on small mammals and insects) which breed July-Nov, and exhibit a strong seasonal pattern in their mode of hunting [27], alternating between hover-hunting (riding thermals, particularly over concrete runways and taxiways where aircraft are moving) and still-hunting (launching from a perch). Nankeen Kestrels are the most frequently struck bird species at Brisbane airport, with strikes throughout the year.

- Cattle Egrets are migratory and have high numbers from November till February (southern summer), with breeding occurring October till January with markedly fewer individuals observed in March–October (southern winter). Cattle Egrets feed on grasshoppers and other insects, foraging on open grassland.

- The Straw-necked Ibis population on/around the airport is the most volatile and inconsistent of the three hazard species, and while numerous, are rarely involved in strike events. These fluctuations cannot be explained by specific criteria such as migratory behaviour, seasonality, or hunting behaviour. Instead, the Straw-necked Ibis population appears to be strongly linked to food availability (leading to birds being attracted to the airport precincts, and increased breeding).

## Methods

The main objective of this study was to investigate a set of data-driven modelling techniques, from the point of view of generating dynamic collision likelihood scores. It was decided to test three (3) separate modelling approaches based, in part, on what has been described in the literature, and in part on novelty of approach, i.e. not described in the bird-strike literature but which has application in related data-driven analyses. The techniques included (i) an algebraic approach based on function approximation, (ii) a probabilistic approach using Bayesian networks, and (iii) an unsupervised machine learning approach utilising K-means clustering. Note that supervised machine learning techniques for modelling likelihood of bird strikes were not considered due to the small number of actual strikes recorded against each hazard

**Table 1. Summary of modelling approaches.**

|  | Algebraic | Bayesian | Clustering |
|---|---|---|---|
| **Model type:** | Deterministic | Probabilistic | Exploratory |
| **Concept:** | Possible to derive a formula that uses (limited) variables to generate a likelihood score | Causal relationships between input variables and prior knowledge (conditional probabilities) can be utilised to derive a likelihood rating | Some 'day types' are frequently associated with strikes. Build groups of day types (clusters) based on similarities and dis-similarities. Identify clusters where strikes occur. |
| **Use:** | Apply 'thresholds' for low, medium, high likelihood | Network 'learns' optimal likelihood categorisation | Attach likelihood scores to individual clusters |
| **Requires:** | • Input variables need to be defined in advance<br>• Tailored to individual hazard species | • conditional probability tables need to be calculated | • Large enough sample of 'days' to build clusters that properly group similar days<br>• Tuning to develop optimal number of clusters |
| **Strengths:** | Limited set of input data values required | Tolerant of missing values and is readily applicable to other species | Clearly determines most indicative variables |
| **Limitations:** | Possibly oversimplified prediction rule—based on a single variable | Requires considerable (local) domain knowledge to translate (subjective) prior beliefs into a mathematically formulated prior | Cannot find non-convex clusters or clusters with unusual shapes (in n-dimensional space) |

species. Table 1 provides a summary of the essential features of each selected modelling approach.

The algebraic (function approximation) approach was to use variables used in Allan's and Paton's approaches, and variables identified in the literature as being indicative of collision risk, to develop an annual risk profile for each of the selected hazard species. The Bayesian network approach was intended to provide a likelihood rating of a strike involving a given hazard species under a given set of environmental conditions. K-means clustering and associated cluster analysis was selected with a view to ascertaining whether it was possible to (i) separate (daily) observations into classes, (ii) to then characterise classes (and hence, species-day observation units) as being high or low risk, and (iii) use cluster analysis as a means of feature reduction/selection (identifying the most indicative variables of the 80 daily measurements).

## Algebraic modelling

This approach is based on the following observations and assumptions.

1. Collisions between birds and aircraft only happen when they are both in the same (air) space at the same time.

2. While being rare in absolute terms (i.e. fewer than 10 strikes per 10,000 flights), strikes happen frequently enough for their occurrence to not be completely random.

3. There exist some combination(s) of (environmental) factors that lead to birds and aeroplanes being in the same space at the same time.

4. Some factors are so-called 'lead' indicators of likelihood of collision. For instance, season is associated with certain usual weather conditions (e.g. rainfall, temperature, hours of daylight, etc.) that influence 'intermediate' indicators such as grass coverage (which in turns influences insect population, etc.). Both lead and intermediate indicators impact on 'immediate' likelihood indicators such as numbers of birds on/near runways (see Fig 3).

5. These factors and the relationships between them can be discovered through data analysis of historical environmental (observation, weather, harassment and strike) data.

Let $E$ be the set of all environmental factors and $e_1, e_2, e_3, \ldots e_n \in E$ be individual environmental factors. Let $H$ be the set of hazard species and $E_h \subseteq E$ be the set of environmental

factors relevant to a given hazard species $h \in H$. Then, there exists some function $f: E \times H \rightarrow$ *likelihood* that maps environmental factors to likelihood of a collision. Further, for $h_1, h_2 \in H$, it is likely that $E_{h_1} \neq E_{h_2}$. That is, different environmental factors are relevant to different hazard species.

Paton, when describing his risk assessment model [18], outlined various likelihood indicators including (i) relative abundance, (ii) frequency of occurrence and or area of occurrence. Carter, in his study on risk assessment and prioritisation of wildlife hazards [19] includes ten likelihood indicators including (i) overall population of the wildlife species (in total number of individuals), (ii) location of the species with respect to flight operations, and (iii) number of reported strikes involving the species. Importantly, in [19], the author also mentions the ability to actually influence the species through wildlife control (harassment) as being relevant to likelihood of involvement in a collision. Lastly, analysis of our own K-means clustering results reveal the significance of the number of individuals belonging to hazard species observed and their proximity to runways (see Results. . .K-Means Clustering).

In this study, it was decided to include (i) a **seasonality** measure, (ii) **abundance** which we define as the number of hazard species counted in the daily count as well as the number of hazard species involved in harassment operations, (iii) **overall count** of wildlife and, (iv) **location** of observed hazard species with respect to flight operations (proximity to runways) as key variables to include as model inputs.

The seasonality measure, based on modular arithmetic, was derived by converting (linear) calendar days to solar days (polar co-ordinates) as follows. For some calendar date $d$ and marker date $m$ representing solar day 0, we define *daydiff(d, m)* as number of days offset of $d$ from $m$. Let $s$ be the seasonality index as:

$$s_m^d = cos(daydiff(d, m)/365)$$

Such an approach means that, if 1-Jan is used as the marker date, days at opposite ends of the calendar year (e.g. in January and December) are seasonally close, while dates in January and June are seasonally distant.

The airfield is divided into eleven wildlife management zones. A location indicator for each zone was derived based on the proximity of the zone from the active runways and approach/take-off pathways. Let $Z$ be the set of zones on the airfield, and $P$ be the set of all proximity measures. Let $Z_p \subseteq Z$ denotes the set of zones which have proximity measure $p$ where $p \in P$.

The abundance of a hazard species was calculated on a daily basis as:

- calculate the count of the individuals belonging to hazard species $h \in H$ on any day $d$ in zone $z \in Z$ is sum of the individuals of the hazard species observed during the daily count plus the individuals of the hazard species harassed by zone;

$$count_h^z = observed_h^z + harassed_h^z \tag{1}$$

- calculate the count of the individuals belonging to hazard species $h \in H$ per proximity area $p \in P$ is sum of the count per zone having the same proximity value;

$$count_h^p = \sum_{z \in Z_p} count_h^z \tag{2}$$

- abundance is the sum of 1/proximity * count per proximity area

$$abundance_h = \sum_{q \in P} \frac{count_h^q}{q} \tag{3}$$

The effect of multiplying by 1/proximity is to give more weight to hazard species that are closer to runways and approach/take-off flight paths. For any hazard species $h \in H$, zone $z \in Z$, and proximity $p \in P$:

We note that (i) non-detection of individuals from the hazard species (either through not being observed in one or any zone during the daily count, or not being involved in any harassment activities on a given day) will result in $count_h^z$ being 0 for each such zone on the airfield, and will be reflected in the $count_h^p$ and $abundance_h$ values for the day, and (ii) overall wildlife abundance on the airfield may be calculated similarly as for a single species.

Paton [18] points to population and proximity of hazard species to runways, and seasonality as factors influencing likelihood of collision. In our approach, we have a single measure, abundance, which combines population and proximity, as well as a seasonality measure. Plotting (seasonality, abundance) pairs across the year gives an abundance distribution and allows us to fit a curve, *expected abundance*, for some hazard species $h$ as $e_h = f(s, a_h)$. In [28, 29] techniques are described for constructive function approximation using Gaussian kernels and sigmoid functions respectively.

The graph of a Gaussian is a characteristic symmetric 'bell curve' shape in which the parameter $a_i$ represents the height of the curve's peak, $b_i$ is the position of the center of the peak, and $c_i$ controls the width of the 'bell'. Such a curve is useful in modelling peaks in population over time.

Gaussian

$$g_i(x) = a_i.exp\left(-\frac{(x-b_i)^2}{2c_i^2}\right) \tag{4}$$

The graph of a sigmoid function has a characteristic 'S' shape. The difference of two sigmoids may be used to cut off a continuous range of the domain to form a 'bump' where the difference between the two sigmoid functions is close to the maximum value of the function over this range, and near the minimum value elsewhere [30]. Such a function is useful in modelling a population that is relatively stable over a period of time.

Difference between two sigmoids resulting in a 'bump'

$$s_i(x) = a_i.\left(\frac{1}{1+exp(-(x-b_i+c_i)k_i)} - \frac{1}{1+exp(-(x-b_i-c_i)k_i)}\right) \tag{5}$$

where the parameter $a_i$ represents the height of the bump (maximum value), $b_i$ is the centre of the bump, $c_i$ controls the width of the bump, and the parameter $k_i$ = controls the steepness of the bump (the range over which the function saturates from minimum to maximum value).

A kernel mix of a set of $m$ gaussian kernels and $n$ sigmoid kernels allows us to model both periods of the year where populations rise to a peak, and periods of the year where populations remain largely constant. Thus, we have the expected abundance function for some hazard

species $h$ at seasonality $s$ as

$$e_h = \sum_{i=1}^{m} g_i(s) + \sum_{i=1}^{n} s_i(s).$$

We then define $l_h^d(s_h^d, abundance_h^d, e_h^d) \rightarrow \{r | r \in \{low, elevated\}\}$ as the collision likelihood for some hazard species $h$ on day $d$. We instantiate $l_h^d$ through logistic regression to model likelihood of a strike involving the respective hazard species on any given day. The trained model was then analysed to determine (i) the importance of the individual inputs, and (ii) the threshold(s) of the attributes that contributed positively to the predicted likelihood. The modelling was done in Python using the *sklearn* library for the LogisticRegression model and coefficient importance, and the *imblearn* library for the SMOTE() method for oversampling of minority class (to deal with unbalanced dataset as strikes were observed only infrequently).

In its simplest form:

$$l_h^d = \begin{cases} low, & \text{if } 0 \leq a_h^d \leq t_a \\ elevated, & \text{if } a_h^d > t_a \end{cases} \tag{6}$$

where $a_h^d \in \{s_h^d, abundance_h^d, e_h^d)\}$ represents the most significant attribute, and $t_a$ represents the threshold for $a_h^d$ found through logistic regression modelling.

## Bayesian networks

Bayesian Networks are a method for graphical representation and probabilistic calculation in uncertain and complex scenarios [31]. Bayesian Networks require (i) a set of random variables, (ii) the conditional relationships that exist between them, and (iii) their probability distributions. A Bayesian network is a directed acyclic graph (DAG) such that each node represents a random variable and has associated probabilistic information. Conditional relationships between variables (nodes) are represented by arcs (edges) joining a node to other nodes.

Thus, Bayesian Networks should have the capability of accounting for all the conditional relationships and uncertainties (such as changes in the number of and types of birds, weather, aircraft movements, runway usage, etc.) as they relate to the likelihood of bird strike making them a good approach for risk assessment. Bayesian approaches have been mentioned in relation to climate and environmental factors influencing forest management [32], for imputing missing values in a data set used for analysis of engine failure following bird strike [33], and for estimating structural damage to aircraft following a bird strike [34]. To date, however, Bayesian Networks have not been applied to assess factors influencing the likelihood of wildlife hazard strike occurrence at airports. To address this gap, we show how Bayesian Networks can be utilised in the context of bird strike likelihood.

Network construction utilised the variables shown in Table 2. Note that as most algorithms for Bayesian Networks do not use continuous valued variables, these were discretised into a maximum of three states ('low', 'medium' and 'high') with ranges and cut-offs based on existing literature in the field, analysis of data, and expert input from the BAC WHM team. The prior knowledge on likelihood of strike is measured and derived from the recorded data. In [18] number of observed, location, and proximity to the runway were the variables that were identified as important factors that impact on the likelihood of a strike. As in our algebraic modelling approach, use location and distance from runway to derive a proximity weighting.

**Table 2. Identified data sources and their description of these variables.**

| Variable/Node | Type | Source |
|---|---|---|
| Season | Discrete | BAC data set |
| Temperature | Continuous | Bureau of Meteorology—AU Government |
| Rain | Continuous | Bureau of Meteorology—AU Government |
| Hunting Behaviour | Discrete | Bird biological data |
| Seasonal Behaviour | Discrete | Bird biological data |
| Observed # Of Species | Integer | BAC dataset |
| Location Risk | Discrete | BAC dataset, Zone (i.e. runway and its vicinity) |
| Aircraft Movement | Continuous | Data driven |
| Zone's population | Continuous | Data driven |
| All population | Continuous | Data driven |
| Likelihood | Discrete | Data driven |

Hence, we derived and calculated the likelihood using observed data as:

$$Likelihood = Location\ proximity\ to\ runway \times No.\ of\ Observed \times Proximity\ weight$$

The Bayesian Networks were developed in GeNIe (https://www.bayesfusion.com/genie/), by integrating all the parameters identified as having direct or/and indirect effect on the occurrence of strikes (see Table 2). Fig 1a–1c respectively show the design configuration of BNM for Nankeen Kestrel, Straw-necked Ibis, and Cattle Egret respectively.

Models were validated using K-fold cross validation (K = 2). Then, the Expectation Maximization (EM) method was used to discover maximum-likelihood estimates for all Conditional Probability Table (CPTs) and for refitting the case file data to the final model while minimizing negative log likelihood. Parameters of the model are re-learned each fold, while the structure of the model remained fixed.

A Receiver Operating Characteristic (ROC) curve can efficiently express the quality of a Bayesian Network model and demonstrate "the theoretical limits of accuracy of the model on one plot" [35]. The ROC curve plots the true positive rate (Sensitivity) against the false positive rate (100-Specificity) considering multiple cut-off points of a parameter. ROC curves are able to express the quality of a model effectively. The ROC curve showcases an insight into the trade-off with selecting any states of a variable/node. The diagonal line on the plot illustrates a baseline ROC curve. In other words, a ROC curve above this diagonal line is a classifier that

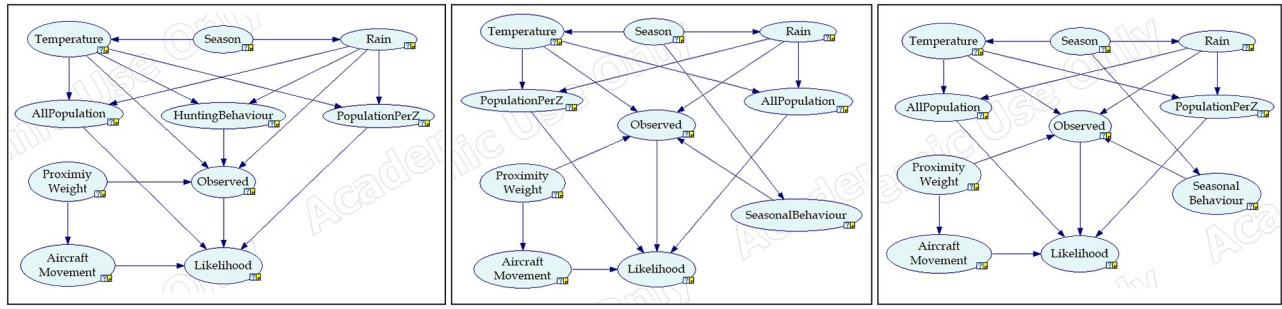

**Fig 1. BNM configuration for Nankeen Kestrel, Straw-necked Ibis, and Cattle Egret.** (a) BNM design for Nankeen Kestrel, (b) BNM design for Straw-necked Ibis, (c) BNM design for Cattle Egret.

demonstrates that the model works effectively. The Area Under the Curve (AUC) is an easy way to identify the quality of the model with a number shown in each ROC diagram. Model quality was assessed by generating ROC curves for each BNM.

**(K-Means) clustering.** The rationale for applying K-means clustering [36] as a data exploration technique was (i) to determine whether airfield days could be clustered into day 'types', i.e. grouped by similarity, (ii) to identify the principal data attributes used to differentiate between day 'types', and (iii) to see whether particular day types are associated with bird strikes.

Accordingly, we took the following steps.

1. Applied unsupervised learning (clustering) to the BAC wildlife observational data to build a 'codebook' of day types. Note that details of strikes were not included in this phase.

2. Validated the models using internal and external validation.

3. Extracted indicative variables using principal component analysis.

4. Used bootstrapping and comparision of collision probabilities at 95% confidence intervals to determine prediction accuracy, i.e., if some day types were actually associated with strikes.

Unsupervised K-means partitions $n$ observations into $K$ clusters such that each observation is assigned to only one cluster, and observations in the same cluster are similar to each other as much as possible, and observations in different clusters are distinctly different. The $K$ centroids serve as prototypes of the respective clusters to which they belong.

Internal validation of clustering may be assessed using silhouette plots. A silhouette plot [37] measures how well each individual observation is assigned to its respective cluster. The y-axis of a silhouette plot shows all the individual data points and the x-axis gives an indication if the data point is farthest away from neighbouring clusters (+1), or is on the decision boundary between two neighbouring clusters (0), or might have been assigned to the wrong cluster (-1). Also, the width the individual silhouette cluster plots and the average silhouette coefficient can give insight if the optimal number of clusters was determined correctly.

Principal Component Analysis (PCA) [38] gives insight into patterns and relationships that exist among the variables, and helps in reducing the complexity of high-dimensional and highly correlated data, while retaining as much of the variance in the dataset as possible. Keeping only the first two principal components finds the two-dimensional plane through the high-dimensional data set in which the data is most spread out. So if the data contains clusters these too may be most spread out, and therefore most visible to be plotted out in a two-dimensional diagram. As a final anlaysis, we applied PCA to identify variables significant in forming clusters.

Thus, in our approach, each cluster will represent a day 'type' and the principal components, identified by PCA, will give insights into the data attributes that differentiate between day types.

Lastly, by calculating the probability of collision on any given day with the probability of collision given that the day belonged to a particular cluster and comparing (95%) confidence intervals for overlap, we determine whether day 'types' are associated with collisions for a particular hazard species. To deal with the low frequency of collisions recorded in the observational data, we use bootstrapping to derive the confidence intervals.

To optimise the data for presentation to the K-means algorithm, we applied data standardisation [39, 40] to address issues of different data ranges, units of measure, and variance apparent in the variables.

Following clustering, internal and external validation measures were applied to determine the goodness of the clustering solution. Internal cluster validation measures reflect on the compactness, connectedness, and the separation of the cluster partitions, and can be determined by the silhouette coefficient (with range [-1..1] [37]) and the Dunn index (with range [0..∞] [41]). The goal is to maximise both the silhouette coefficient the Dunn index [42]. If the data set contains compact and well-separated clusters, the diameter of the clusters is expected to be small and the distance between the clusters is expected to be large (silhouette coefficient will be close to 1 and the Dunn index will be large). That is, we want the average distance within cluster to be as small as possible; and the average distance between clusters to be as large as possible.

The computational environment was the R statistical modelling software version 4.2.0 [43]. The get_clust_tendency() function was used to determine the Hopkins statistic [44] as an indicator of suitability of the data for clustering. The returned (average) value of 0.1, being close to zero, indicated the data was suitable for clustering. The fviz_nbclust() function, using each of the three methods provided for estimating the optimal number of clusters, gave recommended number of clusters, for each species, as two, i.e. K = 2. The kmeans() function was used as the implementation of K-means clustering. The clvalid() function was used to return internal and external measures of validity. The PCA() and get_pca_var() functions were used to return principal components.

## Results

### Algebraic

The expected abundance curves for the three hazard species are shown in Fig 2. Fig 2a) clearly reflects the migratory behaviour of the Cattle Egret. The solar day was zeroed at 1-May meaning values <0 represent the period from November to May. This is the period during which Cattle Egrets begin to arrive in Brisbane (November) and leave Brisbane (late April), with the population peaking in February. This is also the period where the most strikes (orange markers) involving Cattle Egrets were observed. Fig 2b shows the abundance values against solar day (seasonality) for the Nankeen Kestrel. In this case, the solar day (seasonality) value was zeroed at 1-Nov meaning values <0 represent the period from May to November (cooler months of the year with shorter day lengths). Fig 2c shows the abundance values against solar day (seasonality) for the Straw-necked Ibis.

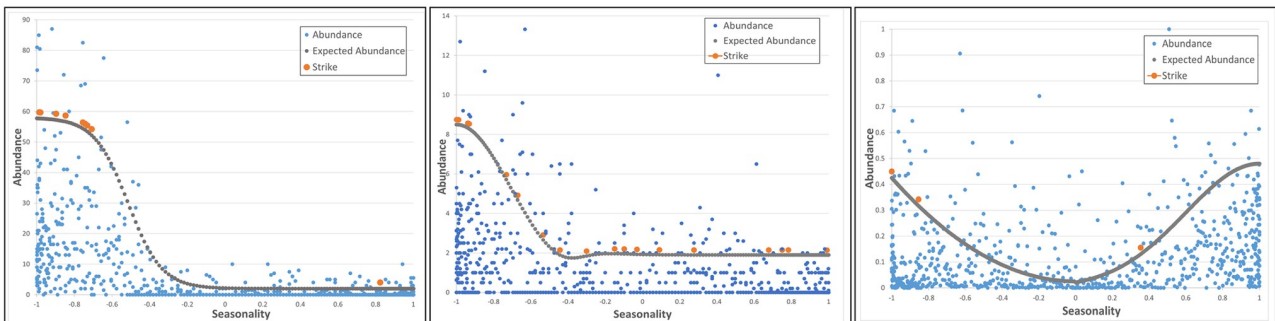

**Fig 2. Hazard species—abundance (vertical axis) against solar day (seasonality) (horizontal axis).** Blue markers indicate abundance on a particular day, orange markers shows days where at least one individual of the relevant hazard species was involved in a strike occurrence, and grey markers give the expected abundance on any day. (a) Cattle Egret abundance, seasonality, and expected abundance, (b) Nankeen Kestrel abundance, seasonality and expected abundance, (c) Straw-necked Ibis abundance, seasonality, and expected abundance.

**Table 3. Logistic regression modelling results for each of the three hazard species showing model quality (accuracy and F Score), attribute with highest feature importance in the model, and threshold value that maximises strike likelihood prediction accuracy.**

| Hazard Species | Accuracy (%) | F Score | Attribute | Threshold |
|---|---|---|---|---|
| Cattle Egret | 86 | 0.86 | Seasonality | -0.78 |
| Straw-necked Ibis | 75 | 0.76 | Abundance | 90 |
| Nankeen Kestrel | $(s_h^d \leq -0.6)$ 64 | 0.61 | Exp Abundance | 7.9 |
|  | $(s_h^d > -0.6)$ 60 | 0.60 | Abundance | 0.1 |

Table 3 shows the results of logistic modelling used to determine the most indicative attribute, and associated threshold value to be used in generating collision likelihood for each hazard species.

Note that the general shape of the expected abundance curve for the Nankeen Kestrel is similar to that of the Cattle Egret, i.e., shows distinct seasonality. Unlike Cattle Egrets, Nankeen Kestrels are involved in collisions throughout the year. We found we got better predictive accuracy by splitting the observational data into two distinct sets at the change in seasonality. Using these attributes and thresholds, the following collision likelihoods may be derived (see Eqs 7–10).

$$\text{Cattle Egret} \quad l_{Egret} = \begin{cases} \text{elevated,} & \text{if } s_{Egret}^d < -0.78 \\ \text{low,} & \text{otherwise} \end{cases} \tag{7}$$

$$\text{Straw} - \text{necked Ibis} \quad l_{Ibis} = \begin{cases} \text{elevated,} & \text{if } abundance_{Ibis}^d > 90 \\ \text{low,} & \text{otherwise} \end{cases} \tag{8}$$

$$\text{Nankeen Kestrel} \quad l_{Kestrel}^{s^d \leq -0.6} = \begin{cases} \text{elevated,} & \text{if } e_{Kestrel}^d > 7.9 \\ \text{low,} & \text{otherwise} \end{cases} \tag{9}$$

$$\text{Nankeen Kestrel} l_{Kestrel}^{s^d > -0.6} = \begin{cases} \text{elevated,} & \text{if } abundance_{Kestrel}^d > 0 \\ \text{low,} & \text{otherwise} \end{cases} \tag{10}$$

The kernel mix parameters for each of the hazard species abundance models are given in S3 Table.

## Bayesian networks

For the Nankeen Kestrel, Straw-necked Ibis and Cattle Egret hazard species, the accuracy of the predicted likelihood is 0.9, 0.96 and 0.85 respectively. The ROC curves for the states of Likelihood (High) are shown in Fig 3.

Figs 4–6 respectively show the BNM's after training for Cattle Egret, Nankeen Kestrel, and Straw-necked Ibis respectively. Each figure highlights the 'influence strength' of connected nodes (shown as arrow width and direction), and includes the conditional probabilities and variables associated with high and low likelihood of collision involving the respective hazard species. For instance, in Fig 4 (i) Season strongly influences Seasonal Behaviour (the Cattle Egret is a migratory species and breeds in summer), and (ii) the likelihood of collision is high when the number of Cattle Egrets Observed is high, with highest collision likelihood when the

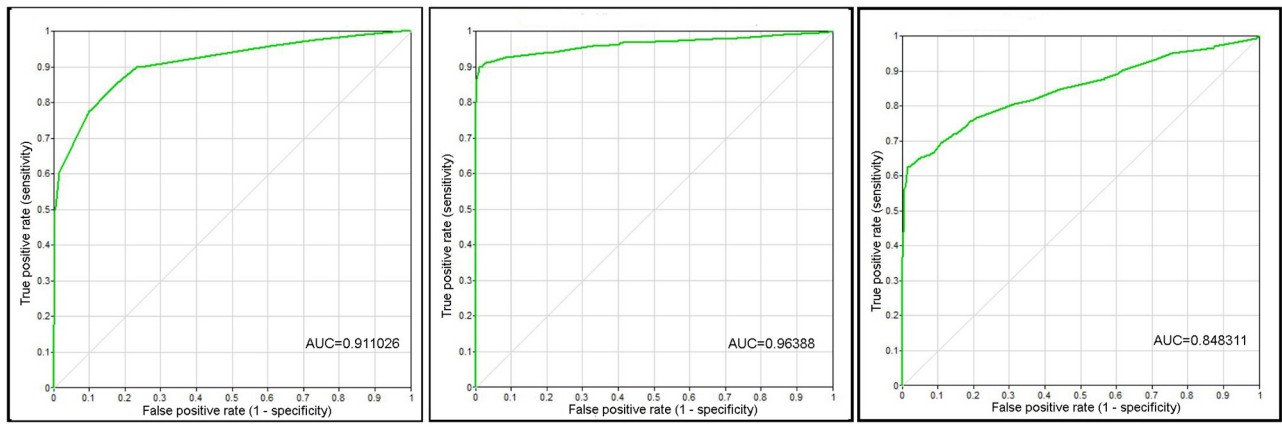

**Fig 3. The receiver-operating characteristic (ROC) curve of the BNM.** (a) Nankeen Kestrel data set, (b) Straw-necked Ibis data set, (c) Cattle Egret data set.

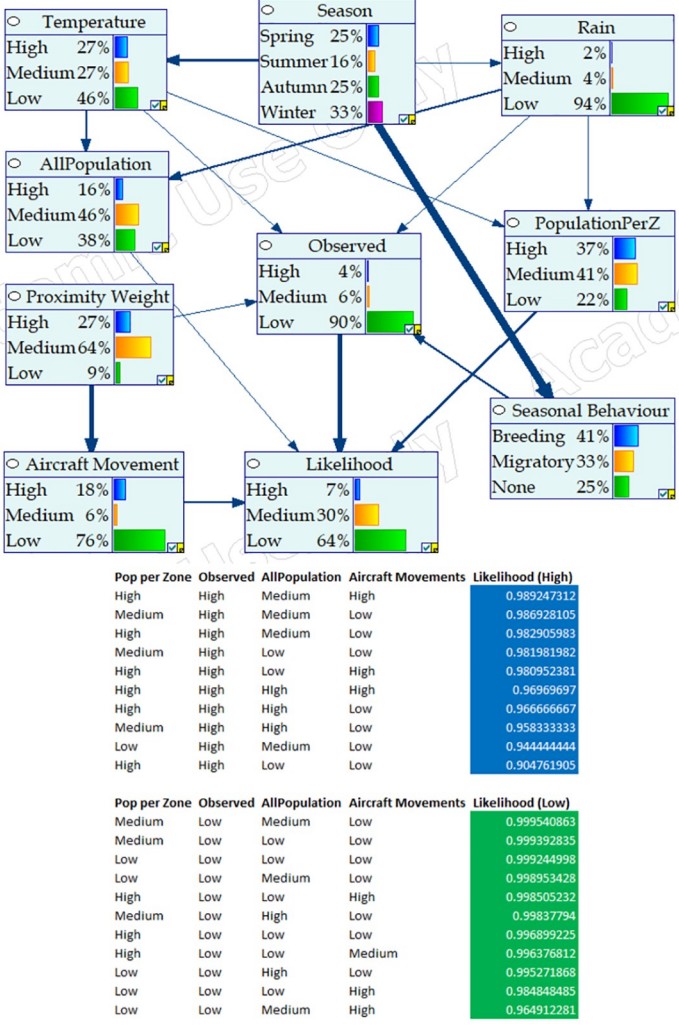

| Pop per Zone | Observed | AllPopulation | Aircraft Movements | Likelihood (High) |
|---|---|---|---|---|
| High | High | Medium | High | 0.989247312 |
| Medium | High | Medium | Low | 0.986928105 |
| High | High | Medium | Low | 0.982905983 |
| Medium | High | Low | Low | 0.981981982 |
| High | High | Low | High | 0.980952381 |
| High | High | HIgh | High | 0.96969697 |
| High | High | High | Low | 0.966666667 |
| Medium | High | High | Low | 0.958333333 |
| Low | High | Medium | Low | 0.944444444 |
| High | High | Low | Low | 0.904761905 |

| Pop per Zone | Observed | AllPopulation | Aircraft Movements | Likelihood (Low) |
|---|---|---|---|---|
| Medium | Low | Medium | Low | 0.999540863 |
| Medium | Low | Low | Low | 0.999392835 |
| Low | Low | Low | Low | 0.999244998 |
| Low | Low | Medium | Low | 0.998953428 |
| High | Low | Low | High | 0.998505232 |
| Medium | Low | High | Low | 0.99837794 |
| High | Low | Low | Low | 0.996899225 |
| High | Low | Low | Medium | 0.996376812 |
| Low | Low | High | Low | 0.995271868 |
| Low | Low | Low | High | 0.984848485 |
| Low | Low | Medium | High | 0.964912281 |

**Fig 4. BNM configuration for Cattle Egret including conditional probabilities and indicative variables for high and low collision likelihood.** Node influence strength is given by the width of connecting arcs.

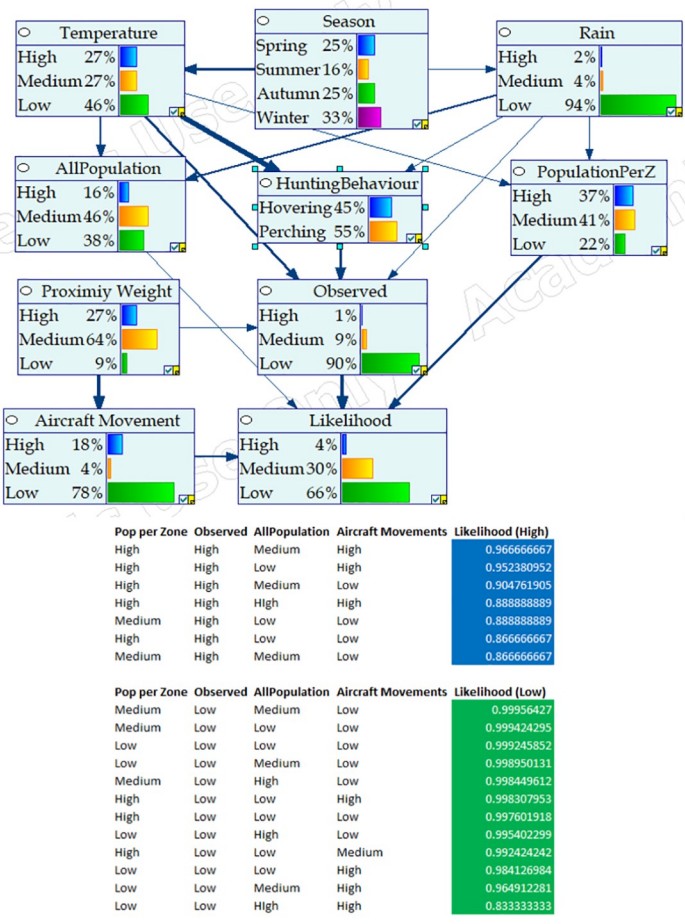

**Fig 5. BNM configuration for Nankeen Kestrel including conditional probabilities and indicative variables for high and low collision likelihood.** Node influence strength is given by the width of connecting arcs.

Population per Zone is High, the number of Cattle Egrets Observed is High, the overall Population of hazard species is Medium, and the number of Aircraft Movements is High.

Conditional probabilities for all nodes, from each of the trained networks are available from our github repository (https://github.com/robertandrews59/ABCBirdstrike).

## K-means clustering

Clustering results are visualised in the 2-dimensional plots in Fig 7 with cluster validity measures shown in Table 4.

Internal validation was by way of silhouette plots. Fig 8 is a silhouette plot for the hazard species Cattle Egret. The 761 observations form two (2) clusters with the average silhouette coefficient 0.33. (The higher the average silhouette coefficient, the more clearly clustering has segmented the data.) Cluster 2 incorporates the majority of the data points (574 out of 761 data points) with 85 points in cluster 1 having a negative silhouette width (avg. -0.11).

Figs 9 and 10 show silhouette plots for the Straw-necked Ibis and Nankeen Kestrel hazard species, again with each plot including the 761 observations of the respective hazard species' data set. For the Straw-necked Ibis, cluster 2 incorporates the majority of the data (570 out of 761 data points) with 83 of the 191 points in cluster 1 having a having a negative silhouette

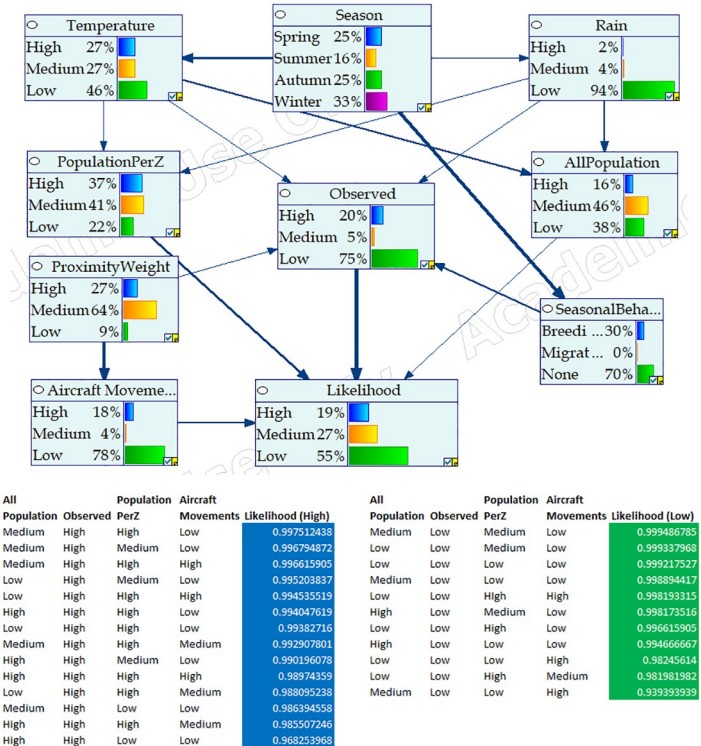

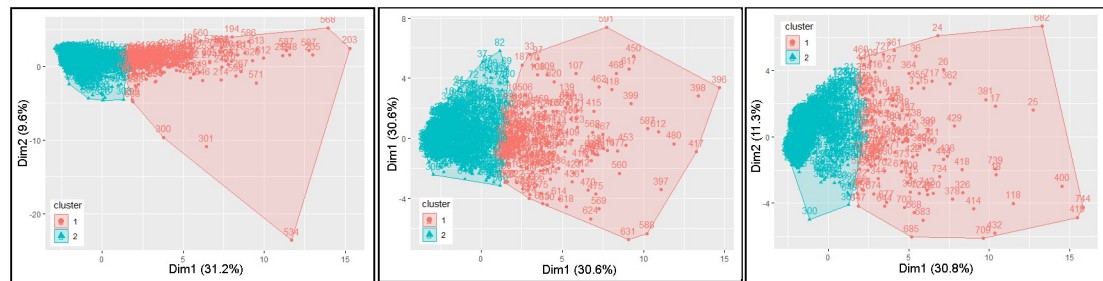

**Fig 6. BNM configuration for Straw-necked Ibis including conditional probabilities and indicative variables for high and low collision likelihood.** Node influence strength is given by the width of connecting arcs.

**Fig 7. 2-dimensional cluster plots for Cattle Egret, Straw-necked Ibis, and Nankeen Kestrel.** The axes for these plots are the principal components of the multi-dimensional vectors used as input to the K-means algorithm. (a) Cluster results for Cattle Egret where k = 2. Cluster 1 includes 187 days, cluster 2 includes 574 days. (b) Cluster results for Straw-necked Ibis where k = 2. Cluster 1 includes 191 days, cluster 2 includes 570 days. (c) Cluster results for Nankeen Kestrel where k = 2. Cluster 1 includes 130 days, cluster 2 includes 631 days.

**Table 4. Cluster validation measures by hazard species.**

| Hazard Species | Clusters | Silhouette Coefficient | Dunn Index |
|---|---|---|---|
| Cattle Egret | 2 | 0.33 | 0.034 |
| Straw-necked Ibis | 2 | 0.28 | 0.109 |
| Nankeen Kestrel | 2 | 0.36 | 0.032 |

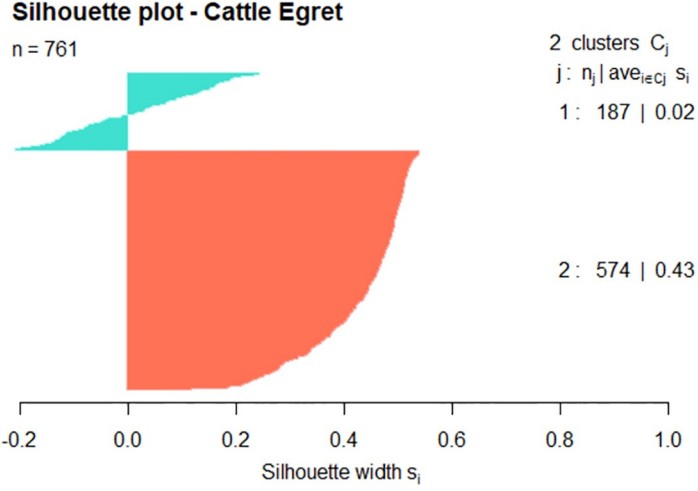

**Fig 8. Silhouette plot of Cattle Egret where k = 2.**

width. For the Nankeen Kestrel, cluster 2 incorporates the majority of the data (631 out of 761 data points) with 57 of the 130 points in cluster 1 having a negative silhouette width.

Analysis of the distribution of bird strikes by species across clusters is shown in Table 5. Analysis is hampered by the low number of strikes observed over the two year period of the study. However, K-means clustering shows that:

- for Cattle Egrets, essentially all strikes occur in day type represented by cluster 1;

- for Straw-necked Ibis, strikes are recorded in both day type clusters; and

- for Nankeen Kestrels, strikes are recorded in both day types, with the majority of strikes occurring in day type represented by cluster 2.

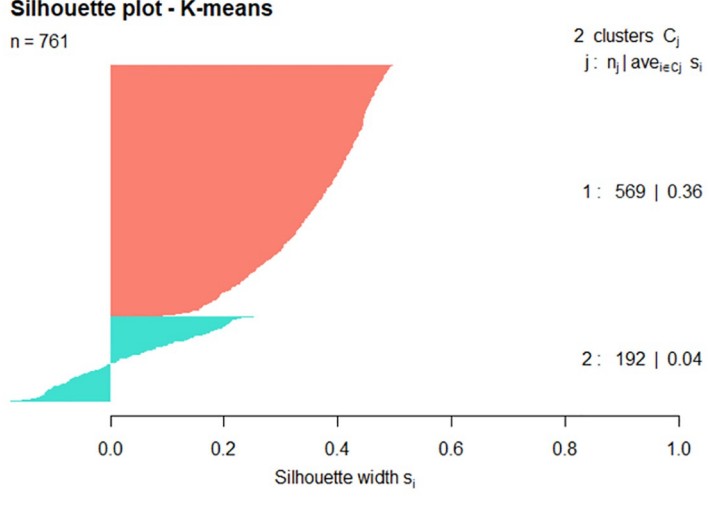

**Fig 9. Silhouette plot of Straw-necked Ibis where k = 2.**

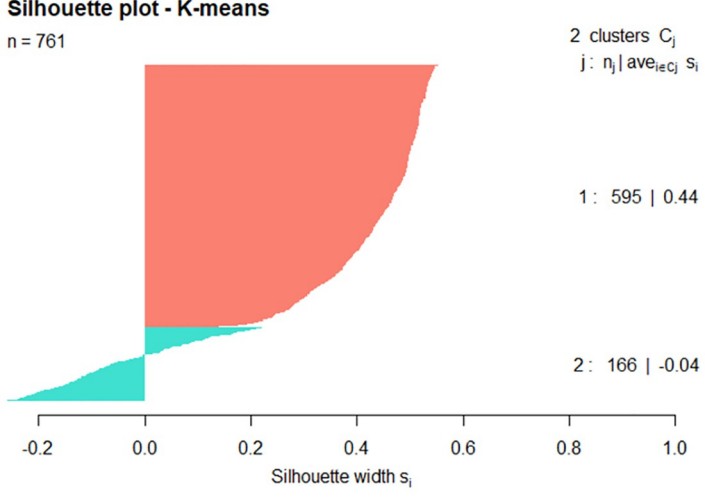

**Fig 10. Silhouette plot of Nankeen Kestrel where k = 2.**

Examination of strike data in S1 Table shows:

- all but one of the strikes involving Cattle Egrets occurred in summer months (Nov-Feb), with a single strike in a winter month;

- of the three strikes recorded for Straw-necked Ibis, two were in summer months and one was in a winter month;

- of the 20 strikes recorded for Nankeen Kestrels, 13 occurred in the months between April and October (cooler months) while 7 occurred in the period December to March (warmer months).

Table 5 also shows, for each of the hazard species, the overall probability of a strike on any given day, the probability of a strike given the day falls into a given cluster, and 95% CI for each of the probabilities. Significance is determined by overlap of the confidence intervals. This analysis indicates that for the hazard species considered, strikes are not associated with day 'types' (as represented by the clusters) in that, apart form one instance, the confidence intervals are not distinct, i.e., they overlap.

**Table 5. Clusters, collision probability, and 95% confidence interval by hazard species.**

| Hazard Species | Cluster | Days | Strikes | P(Collision) | 95% CI | Overlap |
|---|---|---|---|---|---|---|
| Cattle Egret | All | 761 | 9 | 0.012 | [0.005–0.020] | |
| | 1 | 187 | 8 | 0.043 | [0.016–0.075] | Y |
| | 2 | 574 | 1 | 0.001 | [0–0.005] | N |
| Straw-necked Ibis | All | 761 | 3 | 0.003 | [0.0.009] | |
| | 1 | 191 | 1 | 0.005 | [0–0.016] | Y |
| | 2 | 570 | 2 | 0.003 | [0–0.009] | Y |
| Nankeen Kestrel | All | 761 | 20 | 0.026 | [0.016–0.038] | |
| | 1 | 130 | 4 | 0.031 | [0.008–0.061] | Y |
| | 2 | 631 | 16 | 0.025 | [0.015–0.036] | Y |

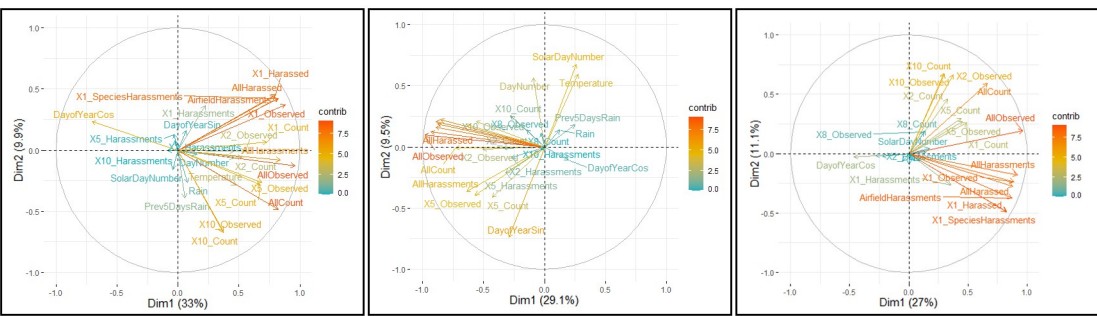

**Fig 11. Hazard species—Principal Component Analysis, PC1 and PC2 for Cattle Egret, Straw-necked Ibis, and Nankeen Kestrel.** For any variable, its contribution to overall variance is given by the distance from the origin (vector length) and is colour-coded (blue indicates a lower contribution, and red the highest). (a) Plot of contribution of variables to PC1 and PC2 for Cattle Egret. (b) Plot of contribution of variables to PC1 and PC2 for Straw-necked Ibis. (c) Plot of contribution of variables to PC1 and PC2 for Nankeen Kestrel.

One of the uses of clustering was to identify variables that contribute to likelihood. Fig 11a–11c respectively show the PCA analysis for Cattle Egret, Straw-necked Ibis, and Nankeen Kestrel respectively highlighting the contributions of individual variables to PC1 and PC2 BNM for each of the three hazard species.

S2 Table lists the two principal components responsible for the largest variation for each of the hazard species. Here, the attributes responsible for the most variation across the three hazard species include attributes related to overall population on the airfield (counted and harassed), population in the zones closest to the runways (counted and harassed), a seasonality indicator (solar day).

## Discussion

Our intent in conducting this work was to develop some techniques to support wildlife hazard risk assessment at airports, that went beyond only hazard ranking, to provide a more dynamic, i.e. sensitive to changing environmental factors, method of providing a collision likelihood. As previously noted, risk assessment also includes taking into account a severity measure. This aspect was outside the scope of this paper, and we concerned ourselves with only collision likelihood and the factors able to be discerned from routinely collected observational data that contribute most to collision likelihood.

A strength of the Algebraic approach is that it clearly visualised the seasonal nature of the abundance of the hazard species included in this study. In particular, the migratory behaviour of the Cattle Egret is evident as is the hunting behaviour of Nankeen Kestrels across the year. For the Cattle Egret, peak abundance corresponds with the period where the most strikes involving Cattle Egrets were observed. For the Nankeen Kestrel we hypothesise that the seasonal abundance is related to the bird's hunting behaviour and the fixed time of the day at which the daily count is conducted, rather than actual changes in the number of birds on the airfield. In summer, the ground over the entire airfield would be warm enough, from late-morning, to generate rising air columns and hence support hover-hunting over most of the airport, particularly over concrete runways and taxi-ways, i.e., spaces where aircraft are departing and arriving. However, in winter months, when the air is cooler, Nankeen Kestrels revert to perch-hunting in the morning (corresponding to the time when the daily count is being conducted) with hover hunting occurring later in the day when the concrete runways and taxi-ways would be warmer than other parts of the airfield, and would thus generate rising air towards the middle of the day. As the daily count is conducted at the same time of day, such

behaviour would affect the number of Nankeen Kestrels observed observed during the daily count.

Expert opinion from the BAC WHM indicated that the population of Nankeen Kestrels on the airfield remained constant throughout the year, yet the abundance figures (derived from the daily count) show fewer birds counted during the warmer period of the year, with more strikes occurring in this period. We ascribe this to the timing of the daily count and the different hunting behaviours exhibited by Nankeen Kestrels at different times of the year. That is, the birds were on/around the airport, but weren't actively hunting or flying at the time the daily count was conducted, i.e., they were perching out of sight and hence were not observed during the daily count. A point worth noting is that, for 7 of the 20 strike events involving Nankeen Kestrels recorded during the study period, there were no Nankeen Kestrels recorded in the daily count or included in harassment activities conducted by the Wildlife Hazard Management team. Although Straw-necked Ibis are among the most abundant bird species on the airport precincts, they are involved in relatively few strikes (3 strike occurrences over the two year period of our study). The BAC WHM report that the species exhibits flocking behaviour, and that it is relatively easy to move the birds to parts of the airport away from runways and taxi-ways. So, although there are large numbers (often several hundreds) of birds on the airport on any given day, they are rarely in close proximity to areas of aircraft movement.

The expected abundance curve however, gives an indication of the expected number of the relevant hazard species at various times of the year and may act as supplement to the daily count of actual abundance. When overlaid with strike information, it provides an easy way to derive rules that quantify collision likelihood. A significant contribution of this approach is the notion of 'seasonality' instantiated in this study as 'solar day'. Seasonality is useful in concentrating routine, daily observations into the seasonality window to make patterns of behaviour more apparent. In this study, a calendar year was used as the seasonality window. This clearly highlighted migratory patterns of the Cattle Egret, and to some extent, the changing hunting behaviour of Nankeen Kestrels at different times of the year. We note that the notion of seasonality may be applied over other time windows. For instance, seasonality could be applied to a daily 24 hour period to highlight diurnal/nocturnal behaviour patterns.

Bayesian Network modelling makes explicit the causal relationships between variables associated with the likelihood of bird strike. Bayesian Networks allow, given, a set of environmental conditions, to reason forward to arrive at a collision likelihood. An additional feature of the Bayesian Network approach is the ability to reason backward from a specified condition (such as high likelihood of collision) to examine the sets of pre-conditions (causal factors) that give rise to the elevated collision likelihood. As such, Bayesian Networks could support decision-making in response to heightened immediate risk, or guide longer-term risk reduction strategies, by identifying currently significant risk factors. Bayesian Networks, in contrast to the simplistic *Risk = Severity × Likelihood*, showcase the integration of all the factors that impact on both elements of risk function (i.e. Severity and likelihood). Bayesian Networks are also able to quantify uncertainty and allow the integration of known probability values associated with any nodes within the networks (prior to an adaptive learning phase). Further, Bayesian Networks provide a visual mechanism to record and test subjective probabilities, an important role in circumstances where there is not much data.

The K-means clustering approach was used to determine whether it was possible to develop a set of 'day types' and then determine if particular 'day types' were associated with strikes. For each of the hazard species considered in this study, (i) clustering generated a set of prototypical 'day types', (ii) strike occurrences (days on which a strike occurred) were localised within particular clusters ('day types'). Principal Component Analysis revealed that numbers of hazard species (counted), their proximity to runways, and seasonality contributed strongly to

clustering of days. We note that these attributes were shown, through logistic regression modelling in the Algebraic approach, to be predictive of elevated collision likelihood. We also note that the seasonality value perhaps subsumes other environmental indicators such as rainfall and temperature as, in Brisbane, these values have distinct ranges associated with the time of year. However, analysis of collision likelihoods for most clusters showed that there was little support for notion of day 'types' determined by clustering being indicative of strikes. We feel that more extensive investigation is required before completely discounting this notion.

As for practical application of the ABC approach to collision likelihood, we feel that the techniques are complementary and should be applied in parallel, and if any one of the techniques predicts a heightened likelihood of collision, actions should be taken. As discussed earlier, the techniques give indications of the factors leading to heightened collision likelihood. These could be used to inform mitigation actions.

The methods described in this paper utilise species-at-a-time modelling. We do not see this as a limitation as it is naive to assume that collisions (strikes) and methods of mitigating strikes are generally applicable to all hazard species and all airports. Our approach provides understanding at the species level, thus allowing for tailored collision likelihood modelling, and potentially tailored mitigation actions. Future work could, however, include generalising the approach to model 'species types'. That is, hazard species could be characterised by a set of features, and risk models constructed for each hazard species type. This would have the advantage of better utilising observational data and strike data as data relating to each species belonging to a particular type would be aggregated to model the type.

The techniques described in this paper can be characterised as having local sphere of application, intermediate forecast horizon, and, at this stage, algorithmic/mode realisation (although the effort required to systematise the individual ABC algorithms would not be great). As such, the ABC approach fills an obvious gap and form a middle ground between purely static risk assessment methods such those described in [11, 18], and immediate indicators such as avian radar making them suitable for application in a wide variety of airports around the world. Neither static, hazard species risk ranking approaches, nor immediate methods, take into account seasonal or environmental factors, hence they cannot be used to (i) reason about likely future states on the airfield, or (ii) to plan mitigation measures in anticipation of some likely future state (a further requirement of airport operators according to CASA and described in [12]). A very obvious example is that of migratory behaviour (as captured in our expected abundance model) or the temporal offset between a rain event and abundance of a given hazard species. That is a) rain, leads to b) grass growing, which provides c) increased food supply for insects and cover for small, ground dwelling animals, which act as d) attractants for insectivorous birds and birds which hunt ground dwelling animals. Such causal chains are nicely captured by the Bayesian Network approach, and can be used at the airport to plan mitigation activities such as mowing, draining of standing water, and removal of perching opportunities for birds across the airport.

## Conclusion

In this paper we have described three techniques (the ABC approach—Algebraic, Bayesian, Clustering) useful in understanding causal factors, and in assessing the likelihood of bird strike at an airport. The techniques were evaluated with routinely collected environment and hazard species data at Brisbane airport.

Contributions of this work to the body of work dealing with aircraft collisions with (avian) hazard species at airports (i) include the definition of attributes for 'seasonality' and 'proximity count' (to deal with risk associated with hazard species at varying distances from aircraft

movement corridors), and (ii) the development of three novel approaches for assessing likelihood of collision.

Each of the three methods improve on simple 'ranking' of hazard species, the minimum required risk assessment for airport accreditation, by providing dynamic risk assessment based on the state of environmental variables. Further, the variables used by the models are those routinely collected by airport Wildlife Hazard Management teams thus implementing the techniques at other airports should not add any burden to data collection, or require changes in data collection practices. Areas for future research around data include (i) how frequently should the historical environmental and collision data be updated in order to keep the models current, i.e., reflective of the actual collision likelihood at the airport, and (ii) what 'window' of historical data is optimal for modelling.

Of particular importance is the identification of variables that contribute strongly to collision likelihood. The Bayesian Network Models are well suited for 'what if' analysis. That is, these models allow for a set of environmental conditions to be provided to the model with the associated risk profile being generated, thus allowing for both diagnostic and prognostic use of the models.

An avenue for future work is to discover means to group 'similar' hazard species and apply the ABC approach to the hazard species group thus making the approach more general. Lastly, the ABC approach fits between purely static risk assessment approaches and purely immediate (e.g. avian radar) in terms of responsiveness to changing conditions on the airfield. The ABC approach also supports planning of mitigation activities through the models providing a 'look ahead' view allowing wildlife hazard management teams to anticipate future risk states. Lastly, this work was developmental. As such, software used was chosen that directly supported the models and provided appropriate analysis and visualisation capabilities. Future development may include development of an integrated platform making all the techniques available in the one tool. Diffusion theory [45] is concerned with the uptake and spread of innovations. It tells us, among others, that technology adoption is driven by not only technological factors, but adoption pathways, time, and the social system around the innovation. In this context, Realisation of the ABC approach as a tool for use in practice has both technological and awareness/acceptance considerations. The technological side will require, among other things, the implementation of each technique in an integrated platform, thus making all the techniques available in the one tool. Such a tool would also require options to allow wildlife control teams to configure routinely collected data for presentation to the models, and to define hazard species relevant to the airport at which the approach was to be deployed. Awareness will require disseminating information about the approach to the industry through for instance wildlife hazard working groups. Acceptance will require a quantum of early adopter airports and their subsequent communication of positive feedback to other airports, leading to more widespread adoption and ultimately approval or even enforcement by from regulatory bodies.

## Supporting information

**S1 Data.**
(ZIP)

**S1 Table. Summary of the strike information for the three hazard species involved in the study during the period of the study.** Dates of strike occurrences are shown together with the number of individuals involved in the occurrence.
(PDF)

**S2 Table. Complete listing of the two principal components, including variable loadings, responsible for the greatest variation as used in clustering models.**
(PDF)

**S3 Table. Configuration parameters for the gaussian and sigmoid functions used in Algebraic models for each of the hazard species, and the 'day 0' value for each of the hazard species.**
(PDF)

## Acknowledgments

We gratefully acknowledge Brisbane Airport Corporation for making members of its Wildlife Hazard Management team available to the research team.

## Author Contributions

**Conceptualization:** Robert Andrews, Moe T. Wynn, Arthur H. M. ter Hofstede, Jackson Ring.

**Data curation:** Robert Andrews, Bayan Bevrani, Brigitte Colin.

**Formal analysis:** Robert Andrews, Bayan Bevrani, Brigitte Colin.

**Funding acquisition:** Moe T. Wynn, Arthur H. M. ter Hofstede.

**Investigation:** Robert Andrews, Bayan Bevrani.

**Methodology:** Robert Andrews.

**Project administration:** Moe T. Wynn.

**Supervision:** Moe T. Wynn.

**Validation:** Brigitte Colin.

**Writing – original draft:** Robert Andrews, Bayan Bevrani, Brigitte Colin.

**Writing – review & editing:** Moe T. Wynn, Arthur H. M. ter Hofstede, Jackson Ring.

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
