## [Decision Letter · Decision Letter 0]

21 Apr 2022

PONE-D-21-25171The ABC of Bird Strikes at BAC: Modelling bird strike likelihood at Brisbane AirportPLOS ONE

Dear Dr. Andrews,

Thank you for submitting your manuscript to PLOS ONE. After careful consideration, we feel that it has merit but does not fully meet PLOS ONE’s publication criteria as it currently stands. Therefore, we invite you to submit a revised version of the manuscript that addresses the points raised during the review process.

We look forward to receiving your revised manuscript.

Kind regards,

Juan Manuel Pérez-García, PhD

Academic Editor

PLOS ONE

Journal Requirements:

2. Thank you for stating the following in the Financial Disclosure section: "Authors receiving funding: MW

Grant: Direct Funding

Funder: Brisbane Airport Corporation

Website: https://www.bne.com.au/corporate

Funder reviewed the manuscript and agreed to publish"

We note that one or more of the authors have an affiliation to the commercial funders of this research study : Brisbane Airport Corporation

3. Please upload a new copy of Figure 7 as the detail is not clear. Please follow the link for more information: https://blogs.plos.org/plos/2019/06/looking-good-tips-for-creating-your-plos-figures-graphics/" https://blogs.plos.org/plos/2019/06/looking-good-tips-for-creating-your-plos-figures-graphics/

Reviewers' comments:

Reviewer's Responses to Questions

**Comments to the Author**

1. Is the manuscript technically sound, and do the data support the conclusions?

Reviewer #1: Partly

Reviewer #2: Partly

2. Has the statistical analysis been performed appropriately and rigorously? 

Reviewer #1: No

Reviewer #2: I Don't Know

3. Have the authors made all data underlying the findings in their manuscript fully available?

Reviewer #1: Yes

Reviewer #2: No

4. Is the manuscript presented in an intelligible fashion and written in standard English?

Reviewer #1: Yes

Reviewer #2: No

5. Review Comments to the Author

Reviewer #1: PONE-D-21-25171

Review:

The MS presents an original approach of determining the dynamic likelihood of bird collisions with aircrafts that has potential application in reducing such incidents to save both human and avian lives. The concept of the MS is interesting, however, its presentation needs major improvement, particularly in terms of explaining/clarifying how the analytical methods were used in this specific problem and how the results were interpreted. Currently, there is large ambiguity and lack of flow in the methods and results that limit the reproduction of this approach. While the ABC approach of dynamic collision risk modeling is elegant, the statistical treatment around it is not robust in the current MS, and the interpretations of whether some ‘days’ have higher collision risk are subjective and not based on rigorous statistical examination. My detailed comments are attached below:

Comments

Introduction

1. The Introduction nowhere acknowledges that bird strike can be a problem for bird populations as well, given the scale of incidents and mortality

2. L 51-70 contains details that are very specific to the case and not relevant for general readers. This section can be substantially revised and reduced keeping in mind the general readership.

3. L 159-184 details of three hazard bird species can be reduced by retaining only the most relevant information.

Study area

4. L 164-170 Authors postulate hypotheses on factors influencing the space use by Nankeen kestrel, but do not return to this point explicitly in the results and discussion section. Results and inferences pertaining to this hypothesis can be added in Discussion as an explanation for seasonal changes in the species’ space use/numbers.

Methods

5. Methods overlap with Results and need substantial revision to improve the flow and clarity. The statistical analysis around the ABC approach can be improved by including objective examination of whether some ‘days’ have higher collision risk (see suggestion below in point 7). Theoretical concepts behind the three methods – Algebraic modelling, Bayesian networks and K-means cluster analysis are explained well, but are sometimes superfluous as these techniques are widely used, and these sections can be reduced by referring to relevant literature wherever applicable, to keep the text tight, to the point, and focused on the application of the technique. More details on how these techniques have been applied such as variables and analytical treatments need to be mentioned explicitly, so that the approach can be clearly understood and replicated. Some analysis that these techniques depend on such as PCA, ROC, and validation techniques for Cluster analysis are not mentioned in Methods and appear abruptly in Results, making it difficult to follow how the methods were applied.

6. L 234 – 320 Algebraic modelling approach has been explained in sufficient detail and clarity to allow replication. However, the content is verbose and can be revised for brevity. For instance, repetition of the mathematical notation in L274 – 276 is redundant, and L283-301 explanations of Gaussian and Sigmoid functions can be reduced.

7. L 304 – 309 The choice of expected abundance thresholds in classifying collision likelihoods as ‘low’, ‘medium’, ‘high’ is subjective and arbitrary. No statistical treatment has been presented that relates expected abundance to air strike probability. An objective way would be to predict collision probability using binomial GLM (logistic regression) by modeling air strike (1/0) on daily expected abundance of hazard species (eh), and to subsequently classify abundance thresholds based on stipulated cut-offs of predicted collision probability.

8. L 310 Bayesian networks have been used to examine causal relationships between factors leading to collisions. Which factors / variables have been examined in this study are not mentioned in Methods.

9. L 316 In Bayesian networks, a node (factor) has a conditional probability distribution table for each parent node (causal factors) that explains the causal relationships. However, these tables are missing from the Results. Only the networks have been shown in Results which tells readers that X leads to Y but not how X influences Y that is vital for this MS – to understand how seasonal environmental changes can lead to conditions promoting collisions. I return to this point in the Results.

10. L 317-333 Content is verbose and can be revised for brevity

11. In (K-means) Clustering, L 338 Although the objective was ‘to see whether particular day types are associated with bird strikes’, the following section on cluster analysis does not explain how this objective is answered from the knowledge of ‘similar days/conditions’ groups.

12. L 339-350 & 370-378 Given that cluster analysis is a very widely used technique, parts of the section can be reduced by citing relevant literature.

13. L 351-359 Data standardization is a widely used technique and can be explained in a single sentence with relevant citation.

Results

Much of the Results are actually details of methods or interpretation of results that should be shifted to Methods and Discussion sections, respectively. Results should provide adequate reporting of the output statistics of each analysis (see detailed suggestions below).

14. Figures lack legends, axis labels and have been presented without diligence that make them difficult to fully comprehend. There are too many figures, and figures of similar type (Fig 1-3, Fig 4-6, Fig 8-10, Fig 11-12, Fig 13-15) can be grouped with species names as labels / icons. Other recommendations specific to each figure are as follows:

15. Fig 1-3: Shift y-axis to the left and remove chart titles.

16. Fig 4-6: These figures show what factors influence collision likelihood. However, the tables/graphs accompanying child nodes seem to be unconditional data distributions that do not allow readers to interpret the nature of the relationships between nodes. Authors can think of presenting the conditional probability tables of child nodes as contingency tables on edges/arrows, or some other concise, interpretable ways, so that readers (including managers) can understand how some environmental changes can promote factors that ultimately lead to higher collision risk.

17. L 381-389 should be removed and included in the figure legend

18. L 390-392 & L 405 & L 415 Collision likelihood based on Algebraic Modeling need to be revised based on the suggestion in point 7.

19. L 397-403 should be included in Discussion and removed from here.

20. L 420-430 details on Bayesian network analysis should be included in Methods and removed from here.

21. L 438 – 448 description of ROC should be included in Methods and removed from here.

22. L 450 – 462 details of Cluster analysis should be shifted from here to Methods

23. L 457 is confusing, as it mentions that PCA can be used as a variable for Algebraic Modeling; however, this variable was not used in Algebraic Modeling and was not referred there. There are several such ambiguous, half-explained statements throughout the MS that makes the methods difficult to follow and replicate.

24. L 463 – 475 details of Cluster analysis validation should be shifted from here to Methods.

25. L 496 Table 4 showing the number of strikes as a function of cluster type is non-informative as we do not know how many days are there in each cluster, to be able to infer if the frequency of strikes was more in one cluster than other. Also cluster difference in collision probability has not been tested statistically, hence the inferences are not reliable. Authors should revise this section (and the corresponding Methods) by including a statistical test of difference in collision probability between day clusters, similar to my suggestion in point 7.

26. L 497 – 511 details of Principle Component Analysis should be shifted from here to Methods. PCA methods should clearly list the variables included and analytical details (data standardization, correlation vs covariance matrix use etc.) in Methods. PCA results should include the variance explained and variable loadings of components in Results.

27. Cluster analysis results should include cluster centroid descriptions in terms of variable means for each hazard species and number of days in each cluster in Results so that readers (including managers) can understand how days are grouped based on conditions and which condition set increases the risk of collision.

Discussion & conclusion

Discussion is weak and largely reiterates the advantages of the approach developed in the MS. It can be strengthened by: a) discussing the findings of applying this approach to the current study in terms of which dynamic factors increased collision risk of a hazard species, and interpreting these results from ecological perspective (see comments in Results section); b) referring to literature on how bird strikes are mitigated across the world beyond simple categorization of hazard species by their risk/severity; and c) how the approach developed in the MS can help advance these current approaches of bird strike mitigation, thereby expanding the scope of the work.

L 556 ‘PCA revealed that number of hazard species … were strong indicators of strikes’ : It is not clear how authors jump to this inference, as Methods do not clarify how daily collision frequency was related to Principle Component(s), and Results do not report any such statistic. The PCA figures 13-15 only show loadings and relative contributions of variables on components, not which component influences collision risk and how. Again refer to my suggestion in point 7 and revise the analytical approach in the MS accordingly.

Conclusion reiterates the approaches used, much of which have already been covered. Instead, it can be reduced to a few important concluding statements on the application of these techniques in reducing bird strike problems.

Reviewer #2: The paper presents three ways to inform collision risk from wild birds. This is an important study that has real world applications. The authors make a case for their study by referencing other risk assessment frameworks and approaches. However being a study that describes novel methods for risk assessment, a solid, readable and fluid methods section is indispensable to the manuscript, which is currently lacking.

There is scope for clarifying description of methods. In particular, the descriptions are presently difficult to follow as the important terms used in the calculations are not ordered. One has to refer to previous pages to understand. The conceptual parts of the 3 methods could be included as they apply to the problem being presented, or skipped altogether. Methods and results sections are intermixed in many ways.

The introduction and discussion are relatively well drafted, however the same clarity does not exist in the methods and results sections.

An accuracy test of the three methods, using predicted collision risk and actual data of collision could be a valuable addition.

Authors may check for the correct use of wildlife ecology terminology viz., population, species, abundance etc. ‘Populations’ are mostly estimates, and not absolute numbers. Moreover non-detection of birds has not been included in the methods, while it has been acknowledged in results (lines 402-403).

1. Not clear if how variation in bird numbers in Paton’s approach is taken into account, and how this approach does not account for seasonal/environmental changes. Authors may elaborate on this.

2. WHM?

3. Study hazard species are those that have a history of collision with aircrafts. However the authors have stated in the introduction that it is important to include species that may not have a history but may still be potential hazards

4. For context, it may be useful to define the main food source (for reference in line 225), habitat requirements/nesting ecology of the hazard species.

5. Para beginning at line 237: for clarity, I suggest the authors provide some context of the references provided here (eg. Carter in his study on risk assessment and prioritisation of wildlife hazards, the author used ….).

6. Line 243 – 245: does this clustering refer to the k-means clustering? The meaning is unclear. Also a reference is probably missing at the end of the sentence.

7. Line 347: number of hazard species should be written as ‘number of individuals belonging to hazard species’.

8. Lines 267 and beyond: the term ‘population’ doesn’t seem appropriate here. Authors could use ‘count per zone’ or ‘individuals per zone’. Note that ‘sum of hazard species .. ‘ and number of individuals of the species (I think the authors mean this) are two separate things.

9. Lines 266 – 271: the language of this explanation is not clear.

10. Equation after line 274: q has not been defined. Do the authors mean p?

11. Line 353-359: standardisation is a standard practice, authors need not explain this.

12. What was done in references 23 and 24 need to be defined here. What is the average 0.1 value, and how is the 0.5 value set as threshold?

13. Lines 364 – 366: describe how.

14. Lines 367 – 369: use this to substantiate on what analyses were done using these packages in r, and which functions were used for which purpose (instead of not defining how average values and recommended number of clusters were derived – ref to comments above).

15. Lines 420 – 429: methodological details which are incorrectly placed in the results section.

6. PLOS authors have the option to publish the peer review history of their article (what does this mean?). If published, this will include your full peer review and any attached files.

Reviewer #1: **Yes: **SUTIRTHA DUTTA

Reviewer #2: No

---

## [Author Response · Author response to Decision Letter 0]

10 Jul 2022

We have provided detailed responses to each comment and suggestion for improvement provided by all of the editor and two reviewers in the Response to Reviewers letter submitted as a file in this upload.

---

## [Decision Letter · Decision Letter 1]

7 Sep 2022

PONE-D-21-25171R1The ABC of Bird Strikes at BAC: Modelling bird strike likelihood at Brisbane AirportPLOS ONE

Dear Dr. Andrews,

Thank you for submitting your manuscript to PLOS ONE. After careful consideration, we feel that it has merit but does not fully meet PLOS ONE’s publication criteria as it currently stands. Therefore, we invite you to submit a revised version of the manuscript that addresses the points raised during the review process.

We look forward to receiving your revised manuscript.

Kind regards,

Juan Manuel Pérez-García, PhD

Academic Editor

PLOS ONE

Journal Requirements:

Additional Editor Comments:

I suggest that you take into consideration the recommendations of both reviewers. In particular reviewer 3 brings two very important issues. The first is the differentiation between "likelihood" and "risk", you should specify what you performed (in my opinion you did likelihood, but not risk). If so, correct in the title and in some parts of the text. Another option is to detail in methods that likelihood was used as a proxy for risk, but it should be adequately justified with references and explaining how you accounting for severity.  

Another critical point indicated by Reviewer 3 is that at no point is it evaluated which of the 3 techniques is better or if it is the combination of them. I consider that including some of this could greatly improve the conclusion of the paper. 

Moreover, I include two additional recommendations. The title is uninformative and could even be misleading due to the presence of so many abbreviations. I understand that a play on words may have been created, but for non-English speaking audiences it is not clear. I suggest avoiding the inclusion of abbreviations (e.g. ABC or BAC) or local names without their country. A suggestion may be “Evaluation of risk assessment techniques to model bird strikes: the case of Brisbane Airport in Australia”. Authors should include the meaning of the abbreviation ABC in the abstract (e.g. in line 39). 

Reviewers' comments:

Reviewer's Responses to Questions

**Comments to the Author**

1. If the authors have adequately addressed your comments raised in a previous round of review and you feel that this manuscript is now acceptable for publication, you may indicate that here to bypass the “Comments to the Author” section, enter your conflict of interest statement in the “Confidential to Editor” section, and submit your "Accept" recommendation.

Reviewer #2: All comments have been addressed

Reviewer #3: (No Response)

2. Is the manuscript technically sound, and do the data support the conclusions?

Reviewer #2: Yes

Reviewer #3: Partly

3. Has the statistical analysis been performed appropriately and rigorously? 

Reviewer #2: Yes

Reviewer #3: Yes

4. Have the authors made all data underlying the findings in their manuscript fully available?

Reviewer #2: Yes

Reviewer #3: Yes

5. Is the manuscript presented in an intelligible fashion and written in standard English?

Reviewer #2: Yes

Reviewer #3: Yes

6. Review Comments to the Author

Reviewer #2: The manuscript is considerably improved, and the authors have addressed all previous comments/suggestions. I recommend few additional suggestions for enhancing the readability and understanding of the general idea of the paper. I would also like to commend the authors on carrying out this important study.

General comments:

- Authors could add a mention of how frequently would the models be updated with emerging information (recent collision information, local weather changes, novel hazard species)?

- Curious why all three methods were carried out in different software, and not one. Authors could add a line justifying this.

- Figure and table titles require editing (typos) and work (addition of information to make it comprehensive by itself, and not dependent on text).

Specific comments:

Abstract

Line 27: and rare bird extinction

Line 29: and bird populations.

Introduction:

Line 161: raptors (“preying on?” small mammals and insects)

Line 177-179: is this a repetition of line 150-153?

Methods:

Line 232: “includes ten likelihood includes ten indicators including” – not clear

Table 2, page 13: Type column for Zone’s population and All population: continuous (instead of continues)

Fig 1, page 14: typos in title (Straw-necked ibis – at two places)

Line 336: demonstrates

Line 390: Connect explanation of silhouette coefficient and Dunn index with subsequent lines (392-395), in terms of which index/coefficient is a measure of what. Moreover these terms do not appear in the results.

Results:

Table 3: Title above table, title needs to be more explanatory (instead of explanation in text)

Line 422: not clear why the split was necessary

Line 429-432: two sentences with repetitive meaning

Line 434: an explanation of one the figures would add value to this section.

k-means clustering; lines 443-453: repetition of methods. Authors could add a few lines “…we found that….”

Lines 454-456: text better suited/more appropriate for figure title

Table 4: title needs revision; cluster tendency is a new term not been mentioned before (in methods)

Figures 8 and 9: reason for placing figures for species as such (1:2)? Figure titles need to explain coefficients etc mentioned in the plot.

Line 460: is average silhouette score the same as silhouette coefficient (mentioned in line 390)? Authors should use the same term throughout the manuscript.

Discussion

Line 568: they are rarely in close to proximity

Line 572: when overlaid with strike information, <it abundance="" expected=""> provides

Line 600: <categorise group=""> days into clusters (?)</categorise></it>

Reviewer #3: Risk assessment procedures, as crucial as they are, do have their limitations in practice as identified by the authors. Hence, I appreciate their approach of investigating new methods to improve the data availability for wildlife controllers in the field.

The authors clearly state in the text that they focus on likelihood, excluding the severity aspect. This is perfectly fine, however, by doing so, they do not create a risk but a likelihood model. This is indicated in the title, but throughout the text, there is still some mentioning of their models as “risk assessment models”. Please correct.

It is interesting to see the three presented approaches as well as their results. What I am missing in the overall discussion is, how they can best be used in practice Lines 43-44 in the abstract state that each of the techniques meet the requirements – so would one be enough? Which one best? Or would a combination be wise after all?

In the conclusion you mention that the ABC approach is useful etc. But what exactly is the approach? Looking at each of the individual models? Combining their outcome? I am missing the big picture here, please enlighten me and the other readers in the discussion or the conclusions section.

In addition, since the authors seem to strive to actually support risk assessment procedures in practice, I am missing an outlook on where they will take the models from here to actually get them to the airports and make them useable for the wildlife control teams.

Comments on the text

• Source 9 (US Wildlife strikes 1990-2015) is outdated. The current version is 1990-2021, please update

• There is a very hard content-related break between the intro and the methods section (l 112-118). Help the reader by ending the introduction /starting the introduction with a brief reiteration of what you are going to present next. E.g. “to develop these models, a case study was performed at Brisbane airport”

• L 138: the text indicates that you selected three out of multiple hazardous species. Is this the case or are these THE three (most) hazardous species? In case it was three out of multiple, please state why you selected those. If it was the three, please clarify

• L 182-183: superfluous, starting the methods with what is written in l 184 is perfectly sufficient

• Table 1: What about limitations? Are these hidden in the requirements? If so, I would relabel the “requirements” descriptor. If not, limitations should be added for the sake of completeness

• L 279: I am missing a source here as well as equation numbers on this page

• Gaußian and Sigmoid functions come out of the blue. Please briefly state what they are and why they are relevant after l 279

• L 315-317: This is a strong statement. Either you have to provide a source or some reasoning why you believe this to back it.

• Equation before Table 2 – how can a likelihood be a function of a risk level? This does not make sense at first sight. Please elaborate and give an explanation what the risk level includes and how it is measured.

• Figures in general (except F8): Text of labels / in bubbles is extremely small and thus almost illegible. Please increase the font sizes. You might want to change color-coding to different shades of grey for readers who print the paper in grey scale (such as myself)

• L 442: please make this a full sentence

• 514-524: Would this information not be more suitable in the intro? And please add some sources for the individual systems/models. Even though in the business for quite a while, I have never heard of the Swiss/Dutch Bird Migration Model. Is this really its name? Please add a reference.

• 532-540: This is interesting but in large parts a repetition of the results – please make that paragraph more concise

• I realize that you discuss the results of the models in different level of details. Why is this? I would recommend to revisit this section and investigate the options to bring more balance into it. In addition, I would expect some mapping to literature and/or observations from BAC or other airports. Do the models reflect reality well?

• L 558-559: the bird’s weren’t active when? On the day when the considered strike occurred? Please clarify

• L 572 it provides (‘it’ is missing)

7. PLOS authors have the option to publish the peer review history of their article (what does this mean?). If published, this will include your full peer review and any attached files.

Reviewer #2: **Yes: **Akanksha Saxena

Reviewer #3: No

---

## [Author Response · Author response to Decision Letter 1]

10 Oct 2022

We have included our detailed responses to reviewers' comments as an uploaded file.

---

## [Editor Report · Decision Letter 2]

4 Nov 2022

Three novel bird strike likelihood modelling techniques: The case of Brisbane Airport, Australia

PONE-D-21-25171R2

Dear Dr. Andrews,

We’re pleased to inform you that your manuscript has been judged scientifically suitable for publication and will be formally accepted for publication once it meets all outstanding technical requirements.

Kind regards,

Juan Manuel Pérez-García, PhD

Academic Editor

PLOS ONE
---

## [Editor Report · Acceptance letter]

28 Nov 2022

PONE-D-21-25171R2 

Three novel bird strike likelihood modelling techniques: The case of Brisbane Airport, Australia 

Dear Dr. Andrews:

I'm pleased to inform you that your manuscript has been deemed suitable for publication in PLOS ONE. Congratulations! Your manuscript is now with our production department. 

Kind regards, 

on behalf of

Dr. Juan Manuel Pérez-García 

Academic Editor

PLOS ONE